# Fungal Laccase Production from Lignocellulosic Agricultural Wastes by Solid-State Fermentation: A Review

**DOI:** 10.3390/microorganisms7120665

**Published:** 2019-12-09

**Authors:** Feng Wang, Ling Xu, Liting Zhao, Zhongyang Ding, Haile Ma, Norman Terry

**Affiliations:** 1School of Food and Biological Engineering, Jiangsu University, Zhenjiang 212013, China; lxu@ujs.edu.cn (L.X.); mhl@ujs.edu.cn (H.M.); 2Institute of Food Physical Processing, Jiangsu University, Zhenjiang 212013, China; 3Key Laboratory of Carbohydrate Chemistry and Biotechnology, Ministry of Education, School of Biotechnology, Jiangnan University, Wuxi 214122, China; 7160201043@vip.jiangnan.edu.cn; 4Jiangsu Provincial Research Center for Bioactive Product Processing Technology, Jiangnan University, Wuxi 214122, China; 5Department of Plant and Microbial Biology, University of California, Berkeley, CA 94720, USA; nterry@berkeley.edu

**Keywords:** laccase, solid-state fermentation, lignocellulosic agricultural wastes, application

## Abstract

Laccases are copper-containing oxidase enzymes found in many fungi. They have received increasing research attention because of their broad substrate specificity and applicability in industrial processes, such as pulp delignification, textile bleaching, phenolic removal, and biosensors. In comparison with traditional submerged fermentation (SF), solid-state fermentation (SSF) is a simpler technique for laccase production and has many advantages, including higher productivity, efficiency, and enzyme stability as well as reduced production costs and environmental pollution. Here, we review recent advances in laccase production technology, with focus on the following areas: (i) Characteristics and advantages of lignocellulosic agricultural wastes used as SSF substrates of laccase production, including detailed suggestions for the selection of lignocellulosic agricultural wastes; (ii) Comparison of fungal laccase production from lignocellulosic substrates by either SSF or SF; (iii) Fungal performance and strain screening in laccase production from lignocellulosic agricultural wastes by SSF; (iv) Applications of laccase production under SSF; and (v) Suggestions and avenues for future studies of laccase production by fungal SSF with lignocellulosic materials and its applications.

## 1. Introduction

Laccases (EC 1.10.3.2) are multicopper oxidoreductase enzymes with the ability to oxidize a broad range of structurally differing substrates (e.g., monophenols, polyphenols, aminophenols, methoxyphenols, aromatic amines) along with the simultaneous reduction of molecular oxygen to water [1,2]. They were first described in *Toxicodendron vernicifluum* (Japanese lacquer tree; formerly called *Rhus vernicifera*) and subsequently in a wide variety of organisms, including bacteria, insects, and fungi (notably, white rot fungi) [3]. Laccases display broad substrate specificity and are applied in many industrial and environmental technology areas, including in textile effluents (decolorization, detoxification), paper production (biobleaching, biopulping), and biopharmaceuticals (transformation of antibiotics, steroids) [4,5,6]. Their ability to remove xenobiotic substances and generate polymeric products makes them useful in bioremediation processes [7,8]. However, their application in biotechnological processes has been limited because of high production costs resulting from low enzyme activity and low yield. Increasing research attention has been paid to effective laccase production strategies associated with increased activity and reduced cost [9,10].

Laccase production is highest for white rot fungi (Basidiomycetes). In the past, submerged fermentation (SF) has been the most commonly used technology for the production of most enzymes, including laccase [11]. SF results in homogeneous distribution of nutrients, which can result in the full contact and absorption of nutrients by cultured microorganisms. However, there has been a trend during the past decade towards the increasing use of solid-state fermentation (SSF) for the production of certain enzymes [7,12]. In SSF, the desired microorganism is grown in the near or complete absence of free water, using an inert or natural substrate as solid support [13,14]. In comparison with SF, SSF more closely simulates the microorganism’s natural environment and has numerous advantages such as being a simpler technique, having lower energy consumption and less pollution as well as higher product recovery [15,16,17,18]. However, the common drawbacks of SSF have also been observed in laccase production, including difficulties in scaling up and large batch-to-batch variation [19,20]. Substrates used for SSF are typically lignocellulosic wastes that contain carbon, nitrogen, and various mineral elements (K, Mg) needed for microorganism growth, enzyme production, and metabolite synthesis [21]. Fungi, particularly white rot fungi, have a strong ability to degrade lignin and cellulosic substances. Many research groups have attempted to improve laccase production by screening fungal strains based on the choice of lignocellulosic waste and optimization of the medium [22,23,24].

Worldwide, ~200 billion tons of agricultural waste are generated each year [25]. Lignocellulose, the major source of agricultural waste, is regarded as a low-cost nutrient substitute for laccase production in SSF systems in comparison with other complex nutrient sources [26]. Lignocelluloses contain three major polymers (cellulose, hemicellulose, lignin) and can be directly depolymerized by laccase as natural substrates [22,27]. Besides serving as a nutrient source, certain lignocellulosic wastes contain natural inductive substances, such as flavonoids and phenolic compounds, which can be applied directly in SSF to enhance fungal laccase production [28,29,30]. The majority of agricultural waste is used as livestock feed, fuel, and in paper production, or (regrettably) burned or left to rot, contributing to environmental pollution and resource waste. Efficient bioconversion of lignocellulose is, thus, an important goal in agricultural waste resource utilization. One review highlighted the potential of lignocellulosic materials to be used in different applications involving biofuels, enzymes, chemicals, pulp and paper, animal feed, and composites [31]. A variety of lignocellulosic residues associated with agriculture (e.g., sunflower seed hulls, sugarcane bagasse, sawdust waste, apple pomace, cotton stalks) have been studied in this regard [20,32,33,34,35,36]. There have been more than 20 review papers published since 2010, and different topics about laccase have been summarized and discussed, including characteristics, expression and regulation, molecular design, production, and applications. Among these publications, Rodriguez-Couto reviewed the production of ligninolytic enzymes by SSF, discussed an SSF bioreactor design for ligninolytic enzyme fermentation, and compared the laccase production of SF and SSF at the reactor scale [19,37]. The physicochemical characteristics and composition of agroindustrial biomass were summarized by Iqbal et al. (2014), where laccase production from lignocellulosic materials and its application in delignification were included [38]. Recently, the regulation of laccase expression and laccase-mediated bioremediation of pharmaceuticals were highlighted by Yang et al. (2017) [39], and the potential of the strategies used for laccase enhancement were updated by Bertrand et al. (2017) [40]. Here, we review the recent status of laccase production technology using lignocellulosic agricultural wastes by fungal SSF, including characteristics and selection of substrates, comparison of SSF and SF in laccase production, screening of fungal strains, and potential applications of laccase production by SSF.

## 2. Laccase Production from Lignocellulosic Agricultural Wastes by SSF

### 2.1. Lignocellulosic Agricultural Wastes

Agricultural waste provides carbon and nitrogen nutrients, which are excellent substrates for fungal growth and laccase production in SSF [41,42]. In general, the most abundant and least expensive lignocellulosic agricultural waste source is crop straw (e.g., from rice, wheat, or corn). Another abundant source (~180 × 10^6^ tons/year worldwide) is bagasse, the dry fibrous material remaining after the extraction of juice from sugarcane [43]. Lignocellulose is composed mainly of cellulose, hemicellulose, and lignin. Lignocellulose composition of agricultural waste varies depending on the source and should be evaluated before the waste is used as a fermentation substrate. The compositions of lignocellulosic wastes commonly used in SSF were summarized by Rodriguez-Couto and Sanromán (2005) [37]. Besides lignocellulose, certain agricultural wastes (e.g., banana skin) are rich in sugars that are also easily metabolized by microorganisms or can be used as a supporting material because of their physical integrity [44]. Therefore, lignocellulosic agricultural waste should first act as the support for the ligninolytic enzyme production, and it can then be a substrate provider for microbes depending on its components. Besides the typical composition of cellulose, hemicellulose, and lignin, the other components, such as sugar, crude protein, and metal ions, should also be analyzed before the lignocellulosic agricultural waste is utilized. This is because those components can affect fungal growth and enzyme production.

Lignocellulose can be degraded by ligninolytic enzymes such as cellulase, hemicellulase, manganese peroxidase (MnP), laccase (Lac), and lignin peroxidase (LiP) [45]. In lignocellulose, cellulose is embedded in hemicellulose and lignin as long fibers. Lignin is a structurally complex aromatic heteropolymer and may promote impermeability by maintaining structure and thereby inhibit utilization of lignocellulose in SSF [46,47]. It was reported that the degradation of cellulose, hemicellulose, and lignin in lignocellulosic agricultural waste was less than 40% after laccase production by SSF [48]. In lignocellulosic agricultural waste, lignocellulose was not the dominant carbon source for laccase production by SSF. Therefore, an exogenous carbon source was often added to the culture medium of laccase SSF, or the other carbon sources in lignocellulosic agricultural waste could also be used as supplementary material.

Laccase-producing microbial strains are able to effectively degrade lignin because they release a powerful extracellular lignin-degrading enzyme system [49]. Lignocellulosic agricultural wastes are valuable substrates for laccase production as they have a high proportion of raw materials and contribute to improved efficiency. There is abundant evidence that lignin stimulates laccase production [50,51]. On the other hand, laccase production may be adversely affected by excessive lignin content. Gómez et al. found that laccase levels in barley bran cultures (1799.6 U/L) were almost 2-fold higher than those in chestnut shell cultures (959.8 U/L) [52]. Chen et al. evaluated residues of seven plant species as substrates for laccase production and observed maximal laccase activity (10,700 IU/g substrate) in rice straw cultures (lignin 10%–15% *w*/*w*) [22]. Laccase activity was also high (7593.3 IU/g substrate) in medium supplemented with water hyacinth (lignin 3.5% *w*/*w*). Moreover, several studies have shown that lignocellulose stimulates laccase production, presumably because of its high cellulose content. Srinivasan et al. reported stimulation of laccase production by cellulose in *Phanerochaete chrysosporium* [53]. Lignocellulosic residues shown to be effective substrates for laccase production in SSF are summarized in Table 1. Lignocellulosic waste is a good candidate for laccase production by SSF because it can function as support, a nutrient source, and as an inducer. 

### 2.2. Supplemental Nutrients

Lignocellulosic wastes are useful substrates in fermentation processes, but supplemental nutrients (carbon and nitrogen sources) are required to promote fungal growth during early stages [3]. Studies of laccase production have generally used defined media [68]. Carbon sources in the medium play an important role in laccase production because they can promote mycelial growth and induce transcription of the laccase gene [69]. In *P. chrysosporium*, ligninolytic gene expression was triggered only by depletion of carbon-based nutrients [68]. Tavares et al. showed that initial glucose concentration was the factor most important for laccase production in *Trametes versicolor*, and that initial concentration of 11 g/L resulted in maximal laccase production (11,403 U/L) [70]. In recent years, there has not been much research on the effect of the carbon source on laccase production because its influence matrix is basically clear. However, the impact of nitrogen sources is highly controversial for the production of ligninolytic enzymes from different organisms [71]. Some strains require excess nitrogen to protect the enzyme, while others are only induced by nitrogen starvation [72]. From some studies, we can also see that organic nitrogen is more conducive to the production of laccase than inorganic nitrogen [33,58]. The carbon/nitrogen ratio is one of the key parameters for laccase production in lignocellulosic wastes. The optimal C/N ratio in laccase SSF varies due to differences in lignocellulosic resources and fungal strains [5,14,62]. Mineral supplements are also necessary for the growth of microorganisms. In particular, elemental phosphorus is crucial because it is part of the backbone of DNA, the carrier and transmitter of genetic information [3]. In the optimization of SSF laccase production by *Coriolopsis caperata*, statistical analysis of eight selected factors showed that KH_2_PO_4_ alone affected the overall production of laccase by 2.84% [3]. The dependence of laccase activity in different fungal species on the carbon and nitrogen source in the medium is shown in Table 2, which indicates that laccase production can be improved by the modification of supplemental nutrients, particularly carbon/nitrogen sources.

### 2.3. Potential Inducers

Certain lignocellulosic residues contain natural inducers that have the potential to enhance laccase productivity, reduce costs, and reduce pollution in SSF. In SSF of *Funalia trogii*, Kudzu vine root, in which flavonoids are the major phenolic compounds, is an effective substrate for laccase production (42.5 IU/g) [15]. In the SSF of *T. versicolor*, oleuropein and hydroxytyrosol from olive leaves acted as major inducers to increase laccase production (276.62 ± 25.67 U/g dry matter) [58]. The other flavonoid-rich agroindustrial residues, such as tata acti green tea leaves, 1% pulp and paper industry effluent (agro based), and 1% wine made from *Syzygium cumini*, have been demonstrated to improve the laccase production by SSF [29]. The phenolic compounds in steam-exploded cornstalk were beneficial for the induction of laccase expression by SSF [48]. Besides the natural inducers present in agricultural wastes, many single inducers have been added to media to increase laccase yield [50]. Aromatic inducers stimulated response signal recognition in white rot fungi, resulting in an intense biological response that induced secondary metabolism, leading to increased laccase concentration [77]. Other compounds, such as Tween 80 and veratryl alcohol, were used to enhance laccase production [34]. Xylidine was reported to increase laccase production more efficiently than copper [3]. The well-studied inducers of increased laccase production are summarized in Table 3. Table 3 shows that copper is an excellent inducer for laccase production when dosed as CuSO_4_ to increase laccase production [78] because laccase is blue copper oxidase, containing four copper atoms per molecule, and the addition of copper may lead to the activation of the metal, resulting in the expression of laccase genes [34]. Unfortunately, most inducers known to date have disadvantages such as toxic effects, high production costs, and environmental pollution. Thus, the search for a natural inducer from the composition of lignocellulosic wastes used as substrates in SSF is a promising approach to address these disadvantages.

To select lignocellulosic wastes for laccase SSF, these key points need to be considered: (i) lignocellulosic wastes should be a suitable support, and potential carbon or nitrogen sources should be available for the fungi if possible; (ii) the composition of lignocellulosic waste, such as the C/N ratio and micro- and macronutrients, should be clear and defined; (iii) any potential inhibitors should be absent from the lignocellulosic waste; (iv) natural inducers are expected to be present in the lignocellulosic waste in addition to the fixed laccase inducer (lignin).

## 3. Comparison of Laccase Production from Lignocellulosic Agricultural Wastes by SSF or SF

Two types of fungal cultivation have been developed: solid-state fermentation (SSF) and submerged fermentation (SF), both of which are commonly used methods for laccase production [82]. However, problems such as low-volume production and high cost remain unsolved, which hinder the wide application of laccase [48]. To solve these problems, researchers studied laccase production by SSF using lignocellulosic agricultural waste as a substrate (Table 1) because SSF is more compatible with the natural growth of the strain and the lignocellulosic substrates are inexpensive, readily available, and environmentally friendly [10,34]. However, it is well-known that scaling up of SSF still has some difficulties, especially for SSF bioreactors. Rodríguez Couto reviewed the development of SSF bioreactors producing laccase and other ligninolytic enzymes and emphasized the necessity of designing new bioreactors or improving existing bioreactors [19]. In that review, it was also mentioned that the SSF reactor has discontinuities and mass transfer limitations for oxygen. To improve the performance of SSF bioreactors, efforts have been made regarding the design of bioreactors. It was reported that a temporary immersion bioreactor was designed for laccase production from *Trametes pubescens* cultivated in sunflower seed hulls under SFF conditions, and no operational problems were detected in the cultivation [54]. However, the use of bioreactors for SSF is still not widespread, and only a few sterile large-scale solid-state bioreactors have been reported in laccase production by SSF. In recent years, many laccases have been produced by SSF using different lignocellulosic wastes as substrates, but few studies have been conducted on the bioreactor culture (Table 1). As shown in Table 4, laccase activity of 2600.33 ± 81.89 U/g was obtained in the steam-exploded cornstalk against 1241.07 ± 70.93 U/g in the untreated cornstalk [48]. Pretreatment of cornstalks with steam explosion increases the pore volume of the substrate, which facilitates nutrient availability for microbial growth and metabolism in the substrate [48,82,83]. Economou et al. (2017) reported good laccase production of 44,363.22 U/g by SSF using spent mushroom substrate in the culture media [62]. However, this result was obtained using a culture in a glass tube, and further scaled-up experiments were suggested for future study. In order to carry out large-scale laccase production by SSF, culture parameters should be investigated at a large scale, and some adjustments are necessary. Moilanen et al. (2014) found a change in laccase activity obtained for a large-scale laboratory culture [63]. Among the culture parameters, the water content is one of the key points for the scaling up of laccase production by SSF. The evaporation of water in a flask vs. bioreactor generally differs, resulting in varied laccase activity obtained at the different culture scales. A high water content in SSF probably resulted in decreased substrate porosity which, in turn, prevented efficient oxygen penetration [7].

Compared to the laccase production under SSF, the application of SF bioreactors seems to be more mature, and laccase production from lignocellulosic materials in SF could be conducted in large-scale bioreactors (Table 4). Lignocellulosic wastes, such as oak sawdust, rice bran, wheat bran, and other food waste, were used as natural, abundant, and cheap sources of nutrients and laccase inducers [85,86,89]. Based on the information in Table 1 and Table 4, the culture period for laccase production in SF is generally shorter than that of SSF. It was indicated that food waste was suitable for laccase production, and the maximum laccase activity reached 54,000 U/L in a 15 L bioreactor after 8 days [89]. Laccase produced by *Pleurotus ostreatus* was significantly increased by the addition of apple pomace, where the maximal laccase production was increased to 114.64 U/mL with 2.5% (*w*/*v*) apple pomace compared with 76.81 U/mL of laccase production without apple pomace addition [93]. In fact, the mass proportion of lignocellulosic materials in SF medium was much lower than that in SSF medium, and their dominant role in SF is as an inducer. Although liquid fermentation with lignocellulosic materials reduces the cost of laccase production to a certain extent and is easier to carry out at a large scale, it does not definitively provide better laccase production (Table 4). In the case of laccase production from *Ganoderma applanatum* with rice bran, a maximal laccase activity of 11,007 U/L was achieved on the 14th day by SF, while a maximum laccase activity of 4000 U/L was observed in SSF [90]. It can be calculated that the laccase productivity was 2.75 U/g/day and 0.286 U/g/day under SF and SSF, respectively, where SF had better performance. However, opposing results were also observed. *Ganoderma lucidum* with rice bran under SF produced 100.13 U/mL of laccase after a 28-day incubation, whereas 156.82 U/g of laccase was obtained with the same culture period under SSF [84]. The laccase productivity was calculated to be 3.57 and 5.6 U/g/day, respectively. The enzyme production and incubation period varied in the laccase production from lignocellulosic raw materials by SF or SSF due to different fungal strains and different lignocellulosic substrates.

## 4. Fungal Strains Effective for Laccase Production from Agricultural Wastes

White rot fungi (species or strains of the phylum Basidiomycota) have a well-documented ability to degrade whole wood with high efficiency and a short fermentation time because they secrete various nonspecific extracellular enzymes (particularly LiP, MnP, and laccase) that break down lignin, cellulose, and hemicellulose [94,95]. There is an increasing research focus on the screening fungal strains suitable for laccase production in SSF.

Laccase production has been reported for numerous genera belonging to the fungal classes Deuteromycetes, Basidiomycetes, Agaricomycetes, and Hyphomycetes (Table 1). On the basis of the enzyme production patterns of an array of white rot fungal strains, Hatakka, in 1994, proposed their division into three categories: (i) lignin-MnP group; (ii) MnP-laccase group; (iii) LiP-laccase group [96]. Kuhar et al., in 2007, proposed the classification of white rot fungi into four groups based on the secretion of (i) laccase and two peroxidases (MnP, LiP); (ii) laccase and one peroxidase; (iii) laccase only; and (iv) peroxidase(s) only [97]. The mechanisms whereby fungi produce these various types of enzymes are unclear and are an important topic for future research.

White rot fungi are the most efficient naturally occurring producers of ligninolytic enzymes [2]. Massive LiP, MnP, and laccase activities were observed during SSF of *Coriolus versicolor* using sweet sorghum bagasse as a substrate [50]. The well-known laccase-producing fungal genera are *Pleurotus* (*P. ostreatus*, *P. pulmonarius*) and *Trametes* (*T. versicolor*, *T. hirsuta*) (Table 1). *T. versicolor* has been intensively studied and commercialized for laccase production. Besides strains of white rot fungi, *Aspergillus niger* is a good producer of all three major ligninolytic enzymes: laccase (9023.67 UI/L), LiP (2,234.75 UI/L), and MnP (8,534.81 UI/L) [64]. Generally, a higher degradation rate of lignocellulose in the SSF support results in a higher yield of ligninolytic enzymes. Therefore, using the degradation rate as an index is helpful for the screening of strains with a high laccase yield. Recently, a simple kinetic model was established to predict temporal fungal enzyme production by SSF on complex substrates, where maximal enzyme activity and incubation time for peak value of enzyme production can be estimated [98]. This strategy can be useful for the screening of fungi producing high yields of laccase under SSF in specific lignocellulosic agricultural wastes and the selection of suitable lignocellulosic substrates by SSF within a certain fungal strain.

Novel fungi producing ligninolytic enzymes were isolated from different sources, such as sea grass, mud, herbaceous weed, and mangrove forests [99,100,101,102]. A marine-derived strain, *Pestalotiopsis* sp. J63, was isolated using a modified medium containing 4 mM guaiacol and exhibited high laccase activity of 10,700 IU/g substrate under SSF [22]. Hariharan and Nambisan isolated 15 fungal strains from dead tree trunks and leaf litters, and *G. lucidum* produced maximal production of ligninolytic enzymes by SSF using pineapple leaves as substrate [103]. Eugenio’s group isolated 127 endophytic fungal strains from *Eucalyptus* trees, and 21 fungal strains possessed the ability of ligninolytic enzyme production, including a member of the family Dothioraceae [78].

Utilization of thermotolerant species may be advantageous for SSF because (i) culture is conducted under non-isothermal conditions, and (ii) the high invasive capacity of mycelia and modification of hyphal morphology under changing temperatures are desirable features in SSF [104]. The genus *Trametes* includes some thermotolerant strains, but few of them have been studied at temperatures above 30 °C. *Trametes trogii* LK13 cultured at 37 °C showed enhancement of laccase activity, mycelial growth rate, thermostability, and tolerance to organic solvents [105]. A temperature increase from 40 to 50 °C was detrimental to the fungus *Fusarium incarnatum* and reduced its laccase production [106]. Higher temperatures may also adversely affect metabolic activities of fungi by denaturing key enzymes [4]. The optimum incubation temperature for *Pycnoporus* sp. SYBC-L1 was 35 °C, which is much higher than those observed for most laccase-producing fungi [107]. For several isolated strains, such as *Neofusicoccum luteum*, *Neofusicoccum australe*, *Hormonema* sp., and *Pringsheimia smilacis*, a maximum temperature of 40 °C was recommended for biotechnological applications involving laccase production [78]. The above strains and others similar to them should be investigated further for the screening of desired laccase production properties, such as thermotolerance, a short fermentation period, and water stress tolerance.

Fungal strains with a high laccase yield from lignocellulosic agricultural wastes is another important key point for low-cost laccase production by SSF. Since the higher degradation rate of lignocellulose results in a higher yield of ligninolytic enzyme, potential laccase high-yield strains may be isolated from the natural site with fast composting of lignocellulosic wastes. Although thermotolerant species can prevent the thermogenesis inhibitory effect on cell growth, low laccase yield may also occur due to the deactivation of laccase resulting from high temperatures. Thus, a thermotolerant strain with a high yield of thermostable laccase can be the target of fungal screening for laccase SSF. At present, natural fungal strains are widely used in laccase production from lignocellulosic wastes by SSF. The construction of genetically modified strains is an alternative way to improve the laccase yield of SSF. Overexpression of homologous or heterologous genes encoding laccase and related ligninolytic enzymes may increase the utilization rate of lignocellulosic wastes and improve laccase production [108]. In addition, heterologous expression of thermostable laccase in a thermotolerant white rot fungi provides good potential for large-scale production of laccase by SSF.

## 5. Application of Laccase Production from Lignocellulosic Wastes by SSF

### 5.1. Lignin Degradation

An obstacle to the utilization of lignocellulosic waste is the stubbornness of lignin, in which cellulose fibrils are embedded, prevent the conversion of structural polysaccharides into fermentable sugars [109]. In recent reports, physical and chemical pretreatments have been reported to degrade lignin into lignocellulosic biomass within a short time (10–40 min) [110,111]. However, these physical or chemical methods have some disadvantages, including high energy consumption and difficult operating conditions [112,113]. Biological pretreatment is regarded as a better strategy for the delignification process of lignocellulosic biomass, which offers a lower energy cost alternative to chemical pretreatment for the reduction in recalcitrance towards cellulolytic enzymes [114,115]. The high efficiency of biological pretreatment was because of the simultaneous production of ligninolytic enzyme systems including LiP, MnP, and laccase [116,117]. Most white rot fungi are used as biological reagents in solid-state fermentation to pretreat lignocellulosic waste, where mushroom production, lignin removal, and enzyme production may be carried out at the same time [118]. The effect of different biological pretreatments on structural components of lignocellulosic wastes is depicted in Table 5. In the SFF of *P. ostreatus* in sugarcane bagasse, the lignin content was reduced from 31.89% to 20.79% after 15 days due to laccase production [33]. A total of 27.83% lignin in cotton stalks was degraded by *Daedalea flavida* under SSF [35]. Lignin degradation of 63% and cellulose enrichment occurred in rice straw treated by *Pyrenophora phaeocomes,* with a final laccase activity of 10,859.51 IU/gds [116]. In addition, a maximum of 74% lignin degradation was observed after 30-day cultivation of *Trichoderma viride* in rice straw after optimizing culture parameters [112].

However, in fungal treatment of lignocellulosic wastes, cellulose degradation is generally accompanied by delignification [48]. To maximize the lignin degradation and selectivity, phenolic supplements were applied in the biological pretreatment of sweet sorghum bagasse by *C. versicolor,* and 45.8% lignin degradation was achieved [50]. In addition to investigating the phenolic compounds as laccase inducers, the potential for metal ions to improve the degradation selectivity in biological pretreatment was also examined. It was reported that Fe^2+^ had a stimulating effect on the lignin degradation of wheat straw by *Trametes gibbosa* and kept the cellulose degradation rate low when the Fe^2+^ concentration was 0.5 mM, providing better selectivity in lignin degradation [119]. To achieve a high lignin breakdown and low cellulolytic enzyme production, the fungal strain *C. versicolor* was selected for biological pretreatment of sweet sorghum bagasse, resulting in excellent cellulose recovery [109].

The biological pretreatment of lignocellulosic biomass by SSF is selective, energy-saving, and effective under mild environmental conditions [114]. Biologically pretreated lignocellulosic materials are widely used in different areas, such as feed, pulp, and energy [90,122]. Sugar yield from the enzymatic hydrolysis of lignocellulosic wastes often increases after biological pretreatment due to lignin degradation. It was reported that enzymatic hydrolysis of pretreated sweet sorghum bagasse resulted in a higher fermentable sugar yield, which was ~2.43 times than that of the control [50]. The treatment by *P. phaeocomes* provided a 4.90-times increase in the sugars released from rice straw after hydrolysis [116]. The glucose yield after enzymatic saccharification of the biologically pretreated cotton stalks was increased more than 2-fold compared with that of the untreated control [35]. The delignification of lignocellulosic materials is important for cellulosic biofuel production, ruminant feed digestibility, and formation of paper products [123,124,125].

### 5.2. Dye Decolorization

Approximate 1 million tons of dye are produced globally each year, and nearly 50% of the dye is discharged into the waste stream or, eventually, into landfill [126]. The treatment of dye-containing wastewater is receiving serious attention worldwide because dyes can cause cancer, mutagenesis, chromosomal fractures, teratogenicity, and respiratory toxicity [32,127]. A variety of physicochemical methods are available for removing dyes, but they are expensive, inefficient, and not suitable for a wide variety of compounds [4]. Biological systems (microbes and produced enzymes) provide an environmentally friendly method for dye decolorization [128]. Among microorganisms, white rot fungi, a group of laccase-producing fungi, are very effective in decomposing synthetic dyes because the structure of dyes resembles the lignin in wood [129,130]. The maximal decolorization rates of 13.6 μmol/h/U laccase for reactive black 5 and 22.68 μmol/h/U laccase for reactive orange 16 were obtained by using the filtrate of *Cyathus bulleri* cultured on wheat bran [126]. The decolorization of malachite green by laccase from *Ganoderma* sp. reduced its toxicity and made it amenable for use in fungal growth [127]. Based on the reported studies, the factors affecting the decolorization of dyes included dye concentration, enzyme dosage, decolorization time, and intermediates, etc. Table 6 shows the effect of laccase produced by SSF on dye decolorization. The degree of decolorization may depend on the reaction rate, which is directly related to the dye’s structure as well as its enzymatic activity and properties [131].

Although enzymatic treatments have many advantages, their decolorization time is generally long due to deactivation of the enzyme. Therefore, the shortcomings of physical, chemical, and biological methods have stimulated interest in developing a combined approach to minimize dye contamination of water and avoid secondary effects [14]. The adsorption process by lignocellulosic materials may be an alternative method for removing dyes from effluents [32]. Several studies have shown the potential for adsorption of different lignocellulosic materials [95,133]. The combination of the adsorption and the biological decolorization by SSF is likely to be a suitable method for dye removal and laccase production. Red 40 dye was adsorbed onto a low-cost waste product, followed by degradation by *T. versicolor* under SSF, and the maximum dye degradation was 96.04% [14]. Besides the above strategy, the lignocellulosic substrates after laccase production by SSF can also be good catalysts and adsorbents for dye decolorization. Since there is a low water content in SSF substrates, the fermented substrates can be dried for future application and stored for a long time. Through this method, part of the produced laccase will be immobilized on the fermented substrates, and the lignocellulosic material became more porous after SSF. Therefore, the stable immobilized laccase on the lignocellulosic support with a high specific surface area will be a good candidate for the treatment of dye wastewater. In our recent research, the dried culture residues from *T. versicolor* under SSF with tea residues exhibited high efficiency and good reusability in dye decolorization [134].

### 5.3. Phenolic Compound Degradation and Others

White rot fungi and laccase have shown great potential in treating phenolic compounds [135]. Bisphenol A (BPA) degradation was found to be accompanied by laccase production, and BPA can induce laccase. During the degradation of BPA by *T. versicolor* under SSF, laccase activity increased rapidly from day 6 to day 10 compared with the untreated control [6]. In addition, laccase-producing fungi were also used to remove other phenolic compounds. Phenol (1 mg/g) was completely degraded by *Penicillium simplicissimum* within 3 days [136]. *Hericium erinaceus* showed 47% total phenol decay associated with laccase and manganese peroxidase activities when it was cultured in olive mill wastewater, and a good mushroom yield was obtained from *H. erinaceus* SSF in olive mill byproducts [137].

Besides the above applications, there are other applications for the SSF of lignocellulosic materials. In fact, other enzymes or target metabolites can be produced simultaneously during the laccase production of SSF from lignocellulosic substrates. In addition to laccase, a bioflocculant and lignocellulase were also obtained in solid-phase-fermented oil palm trunks by *C. versicolor* [57]. Laccase produced from SSF was also used in the synthesis of gold nanoparticles [138].

## 6. Outlook

(1) Lignocellulosic agricultural wastes have been widely used in fungal laccase production by SSF due to their low cost and good prospects. Although different lignocellulosic wastes were tested for laccase production by fungi, the principles for the selection of lignocellulosic substrates are lacking. Since the main components of lignocellulosic substrates (lignin, cellulose, and hemicellulose) can be used as nutrients and inducers for fungal laccase production, the relationship between the main components and laccase production should be established and, at the same time, the mechanism may be revealed. This will be beneficial for the substrate selection of laccase SSF with lignocellulosic agricultural byproducts, where better substrates can be obtained at low cost (fewer selection experiments) and with high efficiency.

(2) Many natural inducers are present in lignocellulosic agricultural wastes. Hence, a possible direction for future research involves the exploration of natural inducers with high induction efficiency in relation to laccase production by SSF. In addition, the induction mechanism should be investigated for the high-efficient inducer, allowing better and cheaper inducers to be developed for the synthesis of laccase by SF or SSF.

(3) The information needed for effective large-scale production of laccase by SSF is still limited. More work is suggested involving the scaling up of fungal laccase production by SSF using lignocellulosic agricultural wastes. From this, the possible problems that exist in laccase SSF will be illustrated, and convincing evidence can be provided for improvement of the bioreactor structure and control system as well as a new design for the reactor used in laccase production by SSF. The above achievements will be fundamental for the future of industrial laccase production from lignocellulosic substrates by fungal SSF.

(4) In the screening of fungal strains used for laccase SSF from lignocellulosic wastes, a few methods have been established for the efficient screening of good laccase producers. However, traditional strain screening is still repetitive and time-consuming work. The method development and application, based on high-throughput screening, can be an alternative for the fungal strain selection with high laccase production by SSF with lignocellulosic wastes. Synthetic biology can also be a useful tool for the construction of thermotolerant strains with the production of thermostable laccase, which will provide good fungal producers of laccase by SSF at a large scale.

(5) Laccase production from lignocellulosic agricultural residues by fungal SSF has exhibited great potential in different application areas. In the biological pretreatment of lignocellulose waste, laccase-producing fungi under SSF are a good strategy. However, delignification efficiency and degradation selectivity are the key points for subsequent applications of lignocellulosic materials. For this purpose, the modification of medium composition has been tested, and good results were obtained. Fungal strains with a high production of thermotolerant laccase are also good candidates for the biological pretreatment of lignocellulose waste. In this case, the pretreatment can be conducted under high temperatures for lignin degradation, resulting in other enzymes becoming deactivated in this process. Deletion of the cellulase gene or inhibition of cellulase expression can also be possible strategies for improve degradation selectivity. Although the process and final product of laccase SSF with lignocellulosic wastes have shown good potential in wastewater treatment, their application is limited at the bench scale. For future studies, it is suggested that the application first be examined at a pilot scale, in addition to finding possible apparatus for its continuous application at a large scale.

## 7. Conclusions

Wide industrial application of laccase has been hampered by problems of high production cost and low yield. Various research approaches have been used in attempts to overcome these problems, e.g., co-culture, biosurfactants, and thermotolerant strains. Use of lignocellulosic wastes (which contain natural inducers) as fermentation substrates is a highly promising approach that reduces costs and environmental pollution. Screening for novel high-yielding strains for laccase production and genetic engineering strategies can also enhance laccase production by fungi under SSF. In addition to using various methods to increase laccase production from lignocellulosic wastes, the scaling up of laccase SSF and the application of laccase produced by SSF are also future research directions.

## Figures and Tables

**Table 1 microorganisms-07-00665-t001:** Laccase production by different white rot fungi grown on different natural supports under solid-state fermentation (SSF) conditions.

Fungus	Support	Cultivation Vessel	Period (Day)	Enzyme Substrate	Laccase Activity	Reference
*Pleurotus ostreatus*	Sugarcane bagasse	Erlenmeyer flask (250 mL)	5	ABTS **	151.6 U/g	[33]
*Ganoderma lucidum*	Rice husks and straw	Polyethylene bags (2 L)	10	ABTS	10.927 U/g	[20]
	Sunflower seed hulls		5	ABTS	16.442 U/g	
*Trametes pubescens*	Sunflower seed	Immersion bioreactor (--)	21	ABTS	4000–6000 UI/L	[54]
*Pseudolagarobasidium acaciicola*	Wheat bran	Erlenmeyer flask (250 mL)	12	ABTS	535,000 U/g	[4]
*Coriolopsis gallica*	Sawdust waste	Erlenmeyer flask (250 mL)	15	2,6-Dimethoxyphenol	4880 U/L	[32]
*Pleurotus ostreatus*	Ammoniated corn straw	Fermentation tray (10.8 L)	20	ABTS	661 U/g	[55]
	Wheat straw	Erlenmeyer flask (250 mL)	9	ABTS	6364 ± 64 U/kg	[56]
*Trametes versicolor*	Oil palm trunk	Erlenmeyer flask (2 L)	28	2,6-Dimethoxyphenol	218.66 U/L	[57]
	Olive leaf	Erlenmeyer flask (250 mL)	12	ABTS	276.62 ± 25.67 U/gds	[58]
	Wheat bran and corn straw	Erlenmeyer flask (250 mL)	10	ABTS	32.09 U/g ds	[6]
	Steam-exploded pretreated cornstalk	Erlenmeyer flask (250 mL)	13	Catechol	2765.81 U/g	[48]
	Corncob waste	Erlenmeyer flask (50 mL)	14	ABTS	8.49 U/gdm	[14]
	Corn silage	Erlenmeyer flask (250 mL)	4	ABTS	180.2 U/L	[59]
	Apple pomace	Erlenmeyer flask (500 mL)	14	ABTS	49.16 ± 4.5 U/gds	[34]
	Pulp and paper solid waste				52.4 ± 2.2 U/gds	
	Alfa fibers				14.26 ± 0.8 U/gds	
	Sugarcane leaves	Erlenmeyer flask (250 mL)	17	Guaiacol	165 U/g	[60]
	Wheat straw				150 U/g	
	Rice straw				145 U/g	
*Pleurotus pulmonarius*	Wheat bran	Erlenmeyer flask (250 mL)	10	ABTS	2860 ± 250 U/L	[61]
	Pineapple peel				2450 ± 230 U/L	
	Bagasse				2100 ± 270 U/L	
	Spent mushroom substrate	Glass tubes (141 mL)	20	Syringaldazine	44,363.22 U/g	[62]
*Cerrena unicolor*	Oat husks	Mixed bench-scale bioreactor (20 L)	19	ABTS	28.2 U/g DM *	[63]
*Aspergillus niger*	Prickly palm cactus husk	Erlenmeyer flask (--)	12	Syringaldazine	9023.67 UI/L	[64]
*Funalia trogii*	Kudzu vine root	Erlenmeyer flask (250 mL)	14	ABTS	42.5 IU/g	[15]
*Daedalea flavida* and *Phlebia radiata*	Cotton stalks	Petri plate (--)	15	ABTS	14.19 ± 0.85 IU/g	[35]
*Coriolus versicolor*	Sweet sorghum bagasse	Erlenmeyer flask (250 mL)	16	ABTS	205.01 ± 10.1 U/g	[50]
*Rhizopus* sp.	Prickly palm cactus husk	Erlenmeyer flask (--)	3	Syringaldazine	1.65 U/g	[65]
*Marasmiellus palmivorus*	Pineapple leaves	Erlenmeyer flask (250 mL)	5	ABTS	667.4 ± 13 IU/mL	[10]
*Pycnoporus sanguineus*	Wheat bran and corncob	Erlenmeyer flask (250 mL)	8	ABTS	138.6 ± 13.2 U/g	[5]
*Trametes versicolor*	Brewer’s spent grain	Erlenmeyer flask (500 mL)	12	ABTS	10,108 ± 157.4 IU/g	[66]
		Plastic tray bioreactor (12 L)			13,506.2±138.2 IU/g	
*Trametes hirsuta*	Pine wood chips/orange peels (1:1)	Rotary drum bioreactor (20 L)	35	ABTS	10498 U/L *	[67]

* The enzyme activity was calculated based on the original data from the reference; ** ABTS: 2,2′-azino-bis-(3-ethylbenzthiazoline-6-sulfonic acid); --: No information of the vessel volume was provided in the reference.

**Table 2 microorganisms-07-00665-t002:** Effect of carbon and nitrogen source on laccase production using lignocellulosic agricultural wastes by SSF.

Species	Support	Carbon/Nitrogen Source	Concentration	Laccase Activity	Reference
*Pleurotus ostreatus*	Dry, ground mandarin peels Grapevine sawdust	Glucose	0.333 g/mL	4.80 ± 0.08 U/L	[73]
		Maltose		6.9 ± 0.4 U/L	
*Pseudolagarobasidium acaciicola*	Decayed wood	Glucose	0.775 g/L	379,000 U/gs	[4]
			0.6625 g/L	535,000 U/gs	
			0.55 g/L	479,000 U/gs	
*P. ostreatus*	Sawdust	Glucose	5 g/L	10.90 ± 0.36 U/g dry wt	[74]
			10 g/L	19.42 ± 0.14 U/g dry wt	
			15 g/L	26.00 ± 0.98 U/g dry wt	
*P. ostreatus*	Sugarcane bagasse	Yeast extract	6.4 g/L	151.6 U/g	[33]
		(NH_4_)_2_SO_4_	2.5 g/L	9.942 U/g	
*Trametes hirsuta*	Kiwifruit	NH_4_Cl	0.150 g/L	1079.8–1139.8 U/L *	[75]
			0.400 g/L	359.9–479.9 U/L *	
			0.600 g/L	839.8–959.8 U/L *	
*P. ostreatus*	*E. benthamii* and bagasse of cassava	(NH_4_)_2_SO_4_	0.111 g/gs	10.11 ± 1.04 U/g ds	[43]
		Saltpetre		13.0 ± 1.29 U/g ds	
		soybean		23.32 ± 2.33 U/g ds	
*Trametes versicolor*	Olive leaves	NH_4_NO_3_	20 g/L	38.47 ± 3.12 U/gds	[58]
		Yeast extract	1%	276.62 ± 25.6 U/gds	
*Daedaleopsis confragosa*	Cherry sawdust	Peptone	0.9 mM	20,204.8 U/L	[76]
*D. tricolor*				16,501.7 U/L	

*: The enzyme activity was calculated based on the original data from the reference.

**Table 3 microorganisms-07-00665-t003:** Inducers applied in laccase production using lignocellulosic agricultural wastes by SSF.

Strain	Support	Inducer	Inducer Concentration	Laccase Activity	Reference
*Pleurotus ostreatus*	Bagasse	CuSO_4_ and	150 µM	86.8 U/g	[79]
		ferulic acid	2 mM	167 U/g	
*Trametes versicolor*	Apple pomace	CuSO_4_	3 mmol/kg ds	49.16 ± 4.5 U/gds	[34]
	Pulp and paper solid waste			52.4 ± 2.2 U/gds	
	Alfa fibers			14.26 ± 0.8 U/gds	
*Pycnoporus sanguineus*	Wheat bran and corncob	CuSO_4_	50 mmol/L	138.6 ± 13.2 U/g	[5]
*Marasmiellus palmivorus*	Pineapple leaves	CuSO_4_	3 mM	627.7 IU/mL	[10]
*Daedalea flavida*	Cotton stalks	Cu^2+^	0.5 mM/g	7.74 ± 0.45 IU/g	[80]
		Gallic acid		6.26 ± 0.55 IU/g	
*Coriolus versicolor*	Sweet Sorghum bagasse	CuSO_4_	2.2 µmol/g	58.2 ± 4.3 U/g	[50]
		Gallic acid	4.4 µmol/g	42.1 ± 3.6 U/g	
		Syringic acid	8.8 µmol/g	67.4 ± 7.7 U/g	
*Pycnoporus sanguineus*	*Eichhornia crassipes* and sawdust	CuSO_4_ and gallic acid	1.5 mM and 40 mM	32.02 U/g ds	[81]
*Trametes versicolor*	Corn silage	CuSO_4_	0.1 mol/dm^3^	1539.4 U/dm^3^	[59]
*Daedaleopsis tricolor*	Cherry sawdust	Veratryl alcohol	0.5% *v*/*v*	27,610.92 U/L	[76]

**Table 4 microorganisms-07-00665-t004:** Maximal laccase activities obtained by different types of cultivation.

Support	Fungus	Scale of Cultivate	Type of Cultivation	Period	Laccase Activity	Reference
Rice bran	*Ganoderma lucidum*	Erlenmeyer flask (--)	SF	28 d	100.13 U/mL	[84]
			SSF		156.82 U/g	
Wheat bran	*Pleurotus ferulae* co-cultured with *yeast*	Erlenmeyer flask (--)	SF	7 d	10,575 U/L	[85]
Oak sawdust	*Trametes versicolor*	Erlenmeyer flask (--)	SF	7 d	0.8 U/mL	[86]
Wheat bran	*Trametes versicolor*	Erlenmeyer flask (250 mL)	SF	7 d	0.93 U/mL	[87]
			SSF	14 d	1.54 U/mL	
Wheat bran	*Cerrena unicolor*	Stirred bioreactor (120 L)	SF	12 d	416.4 U/mL	[82]
Wheat bran	*Cotylidia pannosa*	Erlenmeyer flask (--)	SF	77 h	13 U/mL	[88]
Food waste	*Ganoderma lucidum*	Bioreactor (15 L)	SF	8 d	54,000 U/L	[89]
Wheat straw	*Ganoderma applanatum*	Erlenmeyer flask (100 mL)	SF	14 d	11,007 U/L	[90]
			SSF		4000 U/L	
Wheat bran	*Pleurotus ferulae*	Erlenmeyer flask (--)	SF	7 d	6832.86 U/L	[91]
Synthetic fiber	*Peniophora cinerea* *Trametes versicolor*	Stirred tank bioreactor (1.6 L)	SF	15 d	3500 U/L	[92]
				8 d	75 U/L	

* The enzyme activity was calculated based on the original data from the reference; --: No information of the vessel volume was provided in the reference.

**Table 5 microorganisms-07-00665-t005:** Degradation of lignocellulosic structural components in biological pretreatments.

Support	Fungus	Laccase Activity	Degradation Rate (%, *w*/*w*)			Time (Day)	Reference
			Lignin	Cellulose	Hemicellulose		
Sweet sorghum bagasse	*Coriolus versicolor*	205.01 ± 10.1 U/g	45.80	9.81		20	[50]
Rice straw	*Pyrenophora phaeocomes*	10,859.51 IU/gds	63		51	40	[116]
Wheat bran	*Pleurotus ferulae*	68.9 U/L	62.1	35.6	62.6	19	[119]
Wheat bran	*Ganoderma lucidum*	-- -- --	58.5	-- -- --	-- -- --	-- -- --	[120]
Sugarcane bagasse	*Trametes versicolor*		46				
Rice straw	*Pleurotus ostreatus*		52				
Cotton stalk	*Daedalea flavida* and *Phlebia radiata*	~5 IU/g	35.13	--	--	20	[35]
Hardwoods	*Coniophora puteana* and *Trametes versicolor*	1.54 U/mL	--	30.2–38.7	--	42	[87]
Sugarcane bagasse	*Ceriporiopsis subvermispora*	--	48	--	--	60	[113]
Wheat straw	*Phlebia radiata*	--	9.9	--	--	21	[114]
Wheat straw	*Ganoderma applanatum*	3000 U/L	23.5	10	7	14	[90]
Oak sawdust		1800 U/L	20.5	15.05	13		
Steam-exploded cornstalk	*Trametes versicolor*	2600.33 ± 81.89 U/g	7.8	38.1	27.2	16	[48]
Cornstalk		1241.07 ± 70.93 U/g	4.2	23.2	16.9		
Rice straw	*Trametes viride*	--	74	--	--	30	[112]
Wheat straw	*Chaetomium globosporum*	--	45	--	--	--	[121]
Pearl millet straw		--	48	--	--	--	

--: Not determined or no data provided in the reference.

**Table 6 microorganisms-07-00665-t006:** Application of laccase produced by SSF in dye decolorization.

Strain	Support for SSF	Dye	Dye Concentration (mg/L)	Laccase Concentration	Time (h)	Decolorization Rate (%)	Reference
*Pseudolagarobasidium acaciicola*	Wheat bran	Violet P3P	100	10,000 U/mL 20,000 U/mL 10,000 U/mL	24	97.2	[4]
		Green ME4BL	100			80.3	
		Blue 3R	100			91.3	
*Ganoderma* sp.	Wheat bran	Malachite green	100	30 U/mL	16	~100	[127]
			200		20	~100	
			300		24	72	
			400		28	62	
			500		32	55	
*Coriolopsis gallica*	Sawdust	Reactive black Acid Orange 51	50 50	--	24	67 75	[32]
*Pycnoporus sanguineus*	Wheat bran and corncob	Bromophenol blue	25	5 U/mL	2	90	[5]
		Remazol brilliant blue R	100			80	
		Reactive blue 4	100			60	
*Trametes pubescens*	Sunflower-seed shells	Remazol brilliant blue R	133.33	100 U/L	2	79.4	[54]
*Trametes versicolor*	Brewer’s spent grain	Methyl green	7.5	100 U/L	24	87.7	[66]
		Aniline blue	25			78.48	
*Ganoderma lucidum*	Peach palm	Remazol brilliant blue R	50	53.94 U/L	32	93.97	[132]

--: No data provided in reference.

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
