# Peer review of "Fungal Laccase Production from Lignocellulosic Agricultural Wastes by Solid-State Fermentation: A Review"

_microorganisms, 2019, doi:10.3390/microorganisms7120665_

Round 1

Reviewer 1 Report

The use of ssf for laccase production is an interesting topic and the new research developed in the area makes a review necessary.

I like the information included in the tables.

The major drawback of the article is the English language, I would recommend to use the edition services from the journal or have the article reviewed by a English speaking author.

Author Response

Response to Reviewer 1 Comments

We are grateful for the helpful suggestions from the reviewer. With respect to specific requests for clarification, the following revisions have been made.

Point 1: The use of SSF for laccase production is an interesting topic and the new research developed in the area makes a review necessary. I like the information included in the tables. The major drawback of the article is the English language, I would recommend to use the edition services from the journal or have the article reviewed by an English speaking author.

Response 1: Thanks for your kind suggestion. Based on your comment, the language of this manuscript has been edited by the Specialist editing provided from the MDPI English Editing Service.

Reviewer 2 Report

The processing of agricultural wastes using solid-phase fermentation has been known for more than 50 years, but interest in it has resumed as a method for enzymes production. Therefore this review is modern and relevant, it has been written using references of the most recent years, the material has been fully analyzed and will be useful for a wide range of microbiologists and biotechnologists.

I have a small number of comments:

lines 146-147 instead of “… on the effect of carbon source on laccase, because…”

to write “… on the effect  of carbon source on laccase production, because… “

Lines 156-157 this phrase is incomprehensible, it must be clarified

Table 2 - 4 g/12 mL and 1g/ 9gs are not concentrations

Table 4. Column “Fungus”  is not correct because there are another microorganisms (ex, Lysinibacillus sp.)

Author Response

Response to Reviewer 2 Comments

We are grateful for the helpful suggestions from the reviewer. With respect to specific requests for clarification, the following revisions have been made.

Point 1: The processing of agricultural wastes using solid-phase fermentation has been known for more than 50 years, but interest in it has resumed as a method for enzymes production. Therefore, this review is modern and relevant, it has been written using references of the most recent years, the material has been fully analyzed and will be useful for a wide range of microbiologists and biotechnologists.

Response 1: Thanks for your kind comments.

Point 2: lines 146-147 instead of “… on the effect of carbon source on laccase, because…” to write “… on the effect of carbon source on laccase production, because…”

Response 2: Thanks for your kind suggestion. It has been revised.

Point 3: Lines 156-157 this phrase is incomprehensible, it must be clarified.

Response 3: Thanks for your kind suggestion. This sentence has been rewritten and more information has been added.

Point 4: Table 2 - 4 g/12 mL and 1g/ 9gs are not concentrations.

Response 4: Thanks for your kind suggestion. The concentrations have been calculated and revised in Table 2.

Point 5: Table 4. Column “Fungus” is not correct because there are another microorganisms (ex, Lysinibacillus sp.).

Response 5: Thanks for your kind suggestion. Since this review focused on fungi, the cited content about Rheinheimera sp. and Lysinibacillus sp. has been deleted from Table 4.

Reviewer 3 Report

Microorganisms – review mdpi, Wang et al.

The authors present a synopsis and review current findings related to the production of fungal laccase from lignocellulose containing agricultural wastes using solid-state fermentation. The exhaustive review covers call current and relevant literature comparing types of starting material and fermentation types.

While the review is complete in discussing content related to recent advances in laccase production technology the comments relate mainly to organization and readability.

Abstract:

Line 22 – SSF – run on sentence

Introduction:

Organization – key line 88 – Here….summary of what is presented

English needs re-tooling – examples (not exhaustive list)

Lines 47/51/70/93/107/142/146/178-179/200-201/383/426-427 – tense, plural, not complete sentences.

Lines 321 – In recent literatures – English – i.e. Recent reports

Re-structure Table 1 line 135 – Cultivation vessel – include volume, Period (define – days is assumed but not listed) define enzyme substrate in footnote (ABTS – define in footnote ^ABTS - 2,2′-azino-bis(3-ethylbenzthiazoline-6-sulfonic acid)

Reference needed – line 141

Table 4 – volume for scale of cultivate

Sentence starting line 229 describing table 4 is unclear

Line 322 – define short time

Line 346 – define – It what is “it” referring

Lines 365-366 – dye can cause mutations as they are mutagens – what type of aberrations are the authors describing here, chromosomal?

Author Response

Response to Reviewer 3 Comments

We are grateful for the helpful suggestions from the reviewer. With respect to specific requests for clarification, the following revisions have been made.

Point 1: The authors present a synopsis and review current findings related to the production of fungal laccase from lignocellulose containing agricultural wastes using solid-state fermentation. The exhaustive review covers call current and relevant literature comparing types of starting material and fermentation types. While the review is complete in discussing content related to recent advances in laccase production technology the comments relate mainly to organization and readability.

Response 1: Thanks for your kind comments.

Point 2: Line 22 – SSF – run on sentence.

Response 2: Thanks for your kind suggestion. It has been revised.

Point 3: Organization – key line 88 – Here…. summary of what is presented.

Response 3: Thanks for your kind suggestion. It has been revised and more information has been added.

Point 4: English needs re-tooling – examples (not exhaustive list):

Lines 47/51/70/93/107/142/146/178-179/200-201/383/426-427 – tense, plural, not complete sentences.

Response 4: Thanks for your kind suggestion. All English you commented has been revised. The language of this manuscript has been further edited by the Specialist editing provided from the MDPI English Editing Service.

Point 5: Lines 321 – In recent literatures – English – i.e. Recent reports.

Response 5: Thanks for your kind suggestion. It has been revised.

Point 6: Re-structure Table 1 line 135 – Cultivation vessel – include volume, Period (define – days is assumed but not listed) define enzyme substrate in footnote (ABTS – define in footnote ^ABTS - 2,2′-azino-bis(3-ethylbenzthiazoline-6-sulfonic acid).

Response 6: Thanks for your kind suggestion. Table 1 has been revised according to your suggestion.

Point 7: Reference needed – line 141.

Response 7: Thanks for your kind suggestion. The reference has been added.

Point 8: Table 4 – volume for scale of cultivate.

Response 8: Thanks for your kind suggestion. The cited references in Table 4 have been reviewed and the volume for scale of cultivate has been added in Table 4 if the related information was provided in reference.

Point 9: Sentence starting line 229 describing table 4 is unclear.

Response 9: Thanks for your kind suggestion. The sentence has been revised.

Point 10: Line 322 – define short time.

Response 10: Thanks for your kind suggestion. It has been revised.

Point 11: Line 346 – define – It what is “it” referring.

Response 11: Thanks for your kind suggestion. The sentence has been revised.

Point 12: Lines 365-366 – dye can cause mutations as they are mutagens – what type of aberrations are the authors describing here, chromosomal?

Response 12: Thanks for your kind suggestion. It has been revised and the word “aberrations” has been replaced by “teratogenicity”.

Round 2

Reviewer 1 Report

the new version is fair enough for publication